

**Gradual drying of permafrost peat decreases carbon dioxide in drier peat plateaus but not in wetter fens and bogs**

Aelis Spiller[1], Cynthia M. Kallenbach[2]*, Melanie S. Burnett[1,6,7], David Olefeldt[3], Christopher Schulze[3], Roxane Maranger[4,5], Peter M.J. Douglas[1,6,7]

[1]McGill University, Department of Earth and Planetary Sciences, Montréal, QC, Canada
[2]McGill University, Department of Natural Resource Sciences, Montréal, QC, Canada
[3]University of Alberta, Department of Renewable Resources, Edmonton, AB, Canada
[4]Université de Montréal, Département de sciences biologiques, Complexe des sciences, Montreal, QC, Canada
[5]Groupe de recherche interuniversitaire en limnologie, Montreal, QC, Canada
[6]GEOTOP Research Center, Montréal, QC Canada
[7]Centre d'études nordiques, Québec City, QC Canada

*Correspondence to*: Cynthia.kallenbach (cynthia.kallenbach@mcgill.ca)

**Abstract.** Permafrost thawing of northern peatlands can cause local collapse of peat plateaus into much wetter thermokarst bogs and fens, dominated by *Sphagnum* mosses and graminoids, respectively. However, permafrost thaw can also improve landscape drainage and thus lead to regional drying of peatlands. How gradual drying of these thawing permafrost peatlands
affects the subsequent microbial production of carbon dioxide ($CO_2$) and nitrous oxide ($N_2O$) is uncertain because of landscape heterogeneity in moisture, peat quality, and vegetation. Here, we collected near-surface peat samples (5-20 cm) from Alberta, Canada, across transects representing a thaw gradient from peat plateaus to a fen or bog. We incubated the samples for two weeks at either field moisture conditions or under gradual drying, which reduced moisture by ~80%. Only the fen sites, which had high moisture and % total N, produced $N_2O$ (0.06−6.7 µg $N_2O$-N g$^{-1}$ dry peat) but were unaffected
by the drying treatments. Peat $CO_2$ production was greatest from the fen and the youngest stage of the thermokarst bog despite having the most water-saturated field conditions, likely reflecting their more labile plant inputs and, thus more decomposable peat. We found that $CO_2$ respiration was enhanced by drying in relatively wet sites like the fens and young bog but was suppressed by drying in relatively drier peat plateaus. Further, gradual drying increased $^{13}C$-$CO_2$ respiration, suggesting a possible shift to more decomposed, older C being lost with peat drying. Our study thus suggests that future peat
$CO_2$ and $N_2O$ production from peatlands will depend on whether peat plateaus thaw into fens or bogs and on their diverging responses of peat respiration to more moisture-limited conditions.

## 1 Introduction

Permafrost peatlands are a crucial reservoir of carbon (C), storing around 185 Pg of C (Hugelius et al., 2020). In Canada, 37% of peatlands remain perennially frozen, below a seasonal thawing active layer (Tarnocai et al., 2011), but are
increasingly vulnerable to thaw due to climate change. Following the thaw of permafrost-affected peatlands, the hydrology of the landscape changes, potentially shifting the peat C balance and perturbing the climate system (Varner et al., 2022). As northern permafrost thaws, degradation features develop, including thaw lakes and increased active layer thickness



(Swindles et al., 2015). In response, some areas in the landscape can become water-saturated with ice melt and higher seasonal water tables (Rydin and Jeglum, 2013). However, drier conditions may also occur with thaw depending on the

season or position in the landscape (Boike et al., 2016; Lawrence et al., 2015). For example, temperature-driven increases in evapotranspiration are expected to impart drier conditions on the surface layer of bogs and fens, especially in the summer, and faster drainage may occur with higher hydraulic conductivity following the thaw of underlying ground ice (Lee et al., 2014).

The degree to which permafrost thaw amplifies the release of microbial-produced greenhouse gases (GHGs) into the

atmosphere, exacerbating the global greenhouse effect, will partly depend on whether peat becomes wetter or drier (Kane et al., 2013). Carbon dioxide ($CO_2$) is the dominant respired gas under drier, aerobic conditions, whereas methane, with a comparatively shorter residence time but higher global warming potential, is more prevalent under wetter, anaerobic conditions (Schädel et al., 2016). If a water-saturated ecosystem experiences drying, an increase in more efficient, aerobic microbial metabolism with subsequently higher $CO_2$ production is expected, but this assumes the microbial community will

not become C- or moisture-limited. Recent studies have also demonstrated hot spots of permafrost nitrous oxide ($N_2O$) production with fluctuations between aerobic and anaerobic moisture conditions, challenging past assumptions that permafrost soils lose a negligible amount of nitrogen (N) to $N_2O$ (Ramm et al., 2022; Voigt et al., 2020). Under wetter conditions, $N_2O$ is primarily produced during incomplete denitrification of nitrate. However, anoxic, water-saturated conditions can also inhibit nitrification (and thus nitrate availability), and reduction of $N_2O$ to $N_2$ can be favoured. With drier

or more intermediate moisture conditions, $N_2O$ can be produced via nitrification, so it is possible that a drier permafrost environment can induce $N_2O$ production through enhanced nitrification (Gil et al., 2017).

In heterogenous permafrost environments, where limitations to microbial metabolism can be site-specific, GHG responses to warming or availability of C, nutrients, or $O_2$ is not necessarily similar across the landscape, as previously demonstrated in numerous lab incubations (Laurent et al., 2023; Schädel et al., 2014; Treat et al., 2013). However, to our

knowledge, no study has isolated the effect of gradual drying, often an overlooked result of permafrost thaw, on GHG production across landscape features that vary in their moisture and C and N availability. We conducted an incubation to determine the relationship between experimental short-term gradual drying and the magnitude of $CO_2$ and $N_2O$ gas production across a northern peatland thaw gradient and how drying affects the $CO_2$ isotopic signatures. We interpreted these isotopic signatures as a signal of the C source respired under drier conditions. Methane was not measured since experimental

conditions were not anoxic, required for methanogenesis. We expected that 1) gradual drying will increase $CO_2$ and $N_2O$ production in wetter fens and bogs compared to drier peat plateaus that could become C-limited more rapidly and 2) a decrease in moisture conditions will increase the enrichment of $^{13}C$-$CO_2$, possibly reflecting the metabolism of less decomposed peat.



## 2 Material and Methods

**2.1. Study Site**

The Lutose peatland study site (59.5°N, 117.2°W) is in Alberta, Canada, within the discontinuous permafrost zone (Heffernan et al., 2020). In this region, permafrost peatlands, with 2−6 m deep peat deposits, cover ~40% of the landscape (Vitt et al., 2000). We studied two transects, from a peat plateau to a bog and from a peat plateau to a fen (Fig. 1). These transects represent a space-for-time permafrost thaw gradient with distinct landscape features developing across thaw stages
(Heffernan et al., 2020). For the bog transect, these features include a permafrost peat plateau, a young bog (~30 years since thaw), and a mature bog (~200 years since thaw). The peat plateau has an active layer thickness of ~70 cm, and its surface is raised 1−2 m above the adjacent thermokarst bogs and fens, and has relatively dry surface conditions that deepen with seasonal thaw (Table S1). The young bog (<5 – 10 m wide) is adjacent to the actively thawing area of the peat plateau and has a high growing season water table just below the peat surface. The mature bog, located >10 – 20 m from the thawing
plateau edge, is drier with a lower growing season water table than the young bog (Table S1).

Similarly, in the fen transect, three thaw stages are represented: a permafrost peat plateau, a fen center, and a fen edge. Like the young bog, the fen edge is located where the plateau surface has collapsed. The water table is level with the surface and the peat surface is likely under anaerobic conditions (Heffernan et al., 2024). The fen center also has a high water table but has accumulated new peat material. The fen's relatively wetter conditions and higher pH (Table S1) result from its
additional upstream water sources. In October 2019, we collected four separate peat samples from 5−20 cm at each transect point for a total of 24 cores. Cores were immediately stored on ice, shipped to McGill University, Quebec, and kept at -20 °C until the incubation.

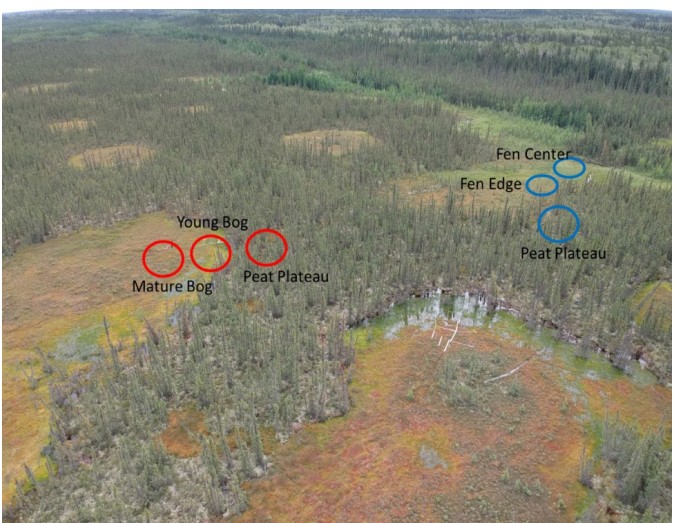

**Figure 1.** The Lutose peatland study site (Alberta, Canada) and the studied bog and fen transects with sample locations
indicated by blue (fen) and red (bog) circles.



## 2.2 Incubation Design for Gradual Drying Experiment

We incubated the thawed (at 4 °C) peat samples from the four replicated cores collected within each transect site: Mature Bog (MB), Young Bog (YB), Peat Plateau 1 (PP1), and Fen Center (FC), Fen Edge (FE), and Peat Plateau 2 (PP2). For each sample, the sample core was homogenized and then $0.52-4.84$ g of dry peat was weighed into a sterile specimen cup and placed in 1 L sealed mason jars with a septum gas sampling port. To create gradual drying conditions for our drying treatment, half of the incubations contained 150 g of calcium sulfate Drierite (>98% $CaSO_4$, <2% $CoCl_2$; 10-20 mesh) outside of the specimen cup that held the peat sample. This approach allowed us to slowly reduce peat moisture content without opening the incubation jars to maintain experimental headspace concentrations. Control incubations at field moisture conditions were maintained similarly but without the Drierite. In addition, we applied the same incubation treatments to a subset of samples to monitor changes in peat moisture during the incubation. We incubated the samples in the dark at 20 °C for two weeks, maintaining the incubation jar seal throughout the 2-week period. Frequent $CO_2$ measurements ensured the jars were maintained below a 3% $CO_2$ threshold of high headspace gas concentrations.

## 2.3 Gas Sampling and Peat Moisture

We sampled headspace $CO_2$ and $N_2O$ at the following hourly time points during the two-week incubation: 0, 6, 24, 48, 96, 168, 240, and 336 h. At each time, 12 mL of gas was sampled with a syringe and stored in evacuated 9 mL vials. The second subset of incubating samples was massed for gravimetric water content at those same time points to produce a peat moisture curve over the incubation period. In addition, at times 0 and 336 h (i.e., the beginning and end of the incubation), we collected 20 mL of gas for isotope analysis stored in evacuated 12 mL vials.

## 2.4 Total and Isotopic Emission Quantification

We analysed total $CO_2$ and $N_2O$ by gas chromatography (Shimadzu GC-2014, Shimadzu Scientific Instruments, Columbia, United States. For $\delta^{13}C$-$CO_2$, we analysed three analytical replicates of 20 mL gas on a Picarro G2201-i Carbon Isotope Analyzer connected to a sample introduction module. We calibrated the instrument using three internal $CO_2$ gas standards (-15.6‰, -28.5‰, and -43.2‰) (Stix et al., 2017).

The $\delta^{13}C$ of the input C source in the headspace gas was calculated with the following mixing model:

$$\delta^{13}C_{input} = \frac{\delta^{13}C_{final} - (\delta^{13}C_{lab\ air} * Fraction\ lab\ air)}{1 - Fraction\ lab\ air}$$

The *fraction of lab air* was calculated by dividing the $CO_2$ concentration at time 0 by that at the time of sampling.



To ensure the $\delta^{13}$C-CO$_2$ of the headspace gas was not an artifact of the Drierite, we ran another two-week incubation with Drierite and without peat, and a control with no Drierite or peat. To simulate the high CO$_2$ concentrations experienced in the original incubations due to peat respiration, we added 60 mL of 5% CO$_2$ gas to the unevacuated Drierite and control jars. Samples from the start and end of the 336 hours were analyzed on the Picarro Carbon Isotope Analyzer. We found that the Drierite had no significant effect on the $^{13}$C isotope composition (mean $\delta$-16.8) compared to the control (mean $\delta$-17.14) (p=0.07), or on the Drierite test CO$_2$ concentrations (mean 1589 ppm) compared to the mean 1523 ppm for the control (p= 0.31). Thus, we ruled out the possibility that changes in incubation $\delta^{13}$C or CO$_2$ ppm values were an artifact of the drying agent.

## 2.5 Peat Chemistry and Isotopic Composition

We measured 1M KCl-extractable nitrate and ammonium colormetrically on all peat samples immediately before and after the incubations. Ammonium and nitrate concentrations in KCl extracts were determined colorimetrically on a 96-well spectrometer (BioTek Synergy Microplate Reader, Santa Clara, CA, USA) with the wavelength set at 650 and 540 nm respectively. We measured peat $\delta^{13}$C and $\delta^{15}$N, total carbon (TC), and total nitrogen (TN) on ground oven-dried peat at the Geotop Stable Isotope Laboratory at the Université du Québec à Montréal using an Elementar Vario MicroCube Elemental Analyzer and, for $\delta^{13}$C and $\delta^{15}$N, coupled with a Micromass model Isoprime 100 isotope ratio mass spectrometer. Results are expressed as $\delta^{13}$C values in ‰ vs. VPDB (Vienna Pee Dee Belemnite) and as $\delta^{15}$N in ‰ vs. AIR (Coplen, 2011).

## 2.6 Statistical Analyses

Means and standard errors were calculated in Matlab and field cores within each transect point were treated as replicates (n=4). Data was checked to ensure homogeneity and normalcy. To test the influence of landscape feature and drying on cumulative gas concentrations, $\delta^{13}$C-CO$_2$, peat C:N, and moisture content, we performed 2-way ANOVA tests, using landscape position and moisture treatment as fixed effects in RStudio (v4.3.2). To compare changes in peat moisture and CO$_2$ and N$_2$O production over time and between moisture treatments, a two-way repeated measures ANOVA was performed within each landscape feature, using the Greenhouse-Geisser test for significant effect. Moisture and time were treated as factors and peat core/incubation jar was the subject of repeated measures. We determined mean differences with Tukey's tests. Lastly, we examined the relationships between changes in peat moisture with drying and CO$_2$ production based on the r$^2$ and linear fit ANOVA p-value from linear regressions. Significance is reported at $\alpha$=0.05.



## 3 Results and Discussion

### 3.1 Initial and Final Peat Moisture

At the beginning of the incubation, peat moisture ranged from 73−95%. Initial field moisture was lowest in the peat plateaus compared to the fen and bogs (p <0.001) (Fig. 2), consistent with other studies from this field site (Heffernan et al., 2024, 2024). In the treatments maintained at field moisture conditions, changes in peat moisture during the two-week period were minimal (Fig. 2b). In the gradual drying treatment, peat moisture declined consistently over time across all the landscape features, except for the peat plateaus which stopped losing moisture at the end of the first week (168 h). Following two weeks of drying, peat moisture was between 10−21%, representing a 71-89% moisture reduction, and was similar across landscape features (p= 0.301).

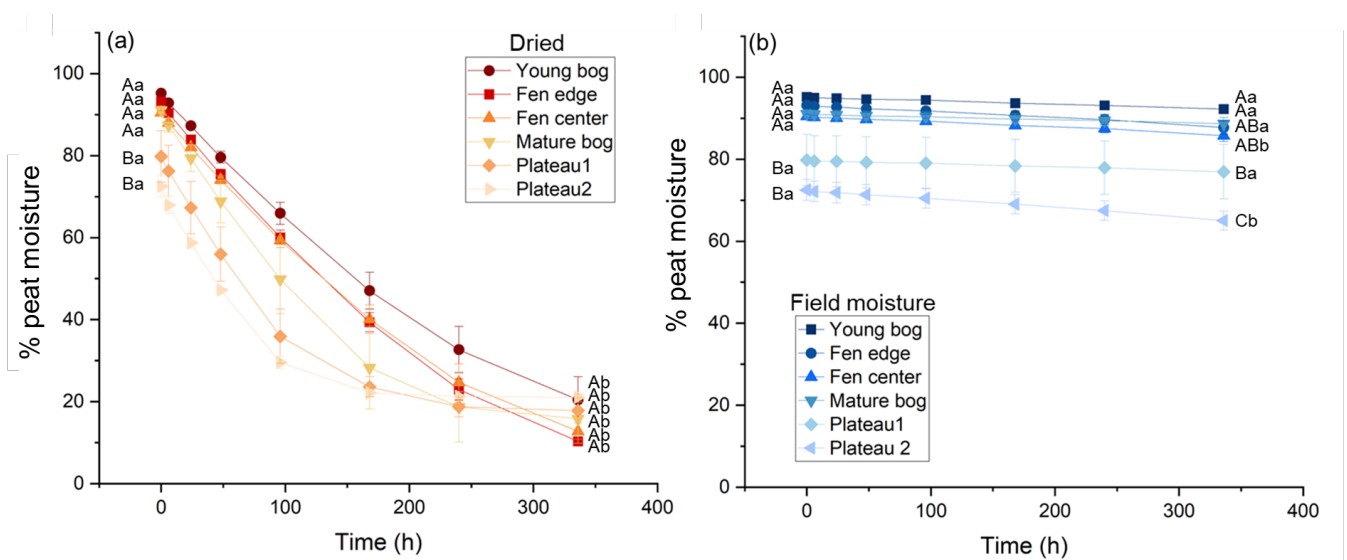

**Figure 2.** Mean peat moisture over incubation time for each landscape feature under gradual drying (a) or maintained at field moisture conditions (b). More color saturation indicates higher *in situ* moisture. Uppercase letters that are different within each panel indicate a significant difference across landscape features for initial and final moisture. Lowercase letters that are different within each panel indicate a significant difference between initial and final moisture within a landscape feature. Error bars are standard error around the mean, n=4.

### 3.2 Landscape Feature Effects on $CO_2$ and $N_2O$

We observed that the YB, followed by the FC and FE, produced the most total $CO_2$-C over the two-week incubation (9.06−14.4 mg g$^{-1}$ dry weight) across moisture treatments (Fig. 3a; Fig. S1). We suspect this higher $CO_2$ production reflects





the more labile plant inputs in the fen and young bog, and thus more decomposable peat (Table S1) that can support elevated

microbial respiration, despite the potential for lower oxygen availability under the field moisture conditions.

We only detected $N_2O$ emissions from the fen center and fen edge, and these were highly variable (0.03−8.3 µg $N_2O$-N $g^{-1}$ dry peat $h^{-1}$), with no effect of drying (Fig. 3b; Fig. S2). This is in contrast to a recent review that showed lower $N_2O$ production in wetland permafrost compared to upland permafrost sites (e.g. peat plateaus), which they attributed to a combination of inhibited nitrification and more complete denitrification to $N_2$ (Voigt et al., 2020). However, in our sites,

initial nitrate concentrations indicate nitrification potential and we do not expect that our incubations were either water-saturated or N-limited enough to cause complete denitrification during the incubation. While the sedge-dominated fens had similar concentrations of inorganic N concentrations to the bogs, they had the lowest C:N (<25) and the highest %TN (Table S1). This may contribute to rapid N cycling and thus higher $N_2O$ production during the incubation, whereas the *sphagnum*-dominated bogs are potentially more N-limited based on the observed higher C:N and more depleted $^{15}N$ (Klemedtsson et al.,

2005; Liao et al., 2021). The peat plateaus and bogs at the same study site were previously found to be $N_2O$ sinks (e.g. consumption of $N_2O$ to $N_2$) *in situ* where anaerobic conditions occur (Schulze et al., 2023). Our observed $N_2O$ emissions from the fen center are considerably elevated for this region, and within the range of many temperate fertilized ecosystems (Elberling et al., 2010). These results further support the emerging evidence that permafrost ecosystems have $N_2O$ hotspots that should be accounted for in estimating the global warming potential of GHG production.

**3.3 Effects of Drying on $CO_2$ Production**

We observed an effect of gradual drying on cumulative $CO_2$ production within each site, though it did not consistently impart a negative or positive forcing on $CO_2$ production. Rather, drying reduced $CO_2$ in the peat plateaus but marginally increased total $CO_2$ production for the YB, FC, and FE sites (Fig. 3a) (interaction p <0.008).

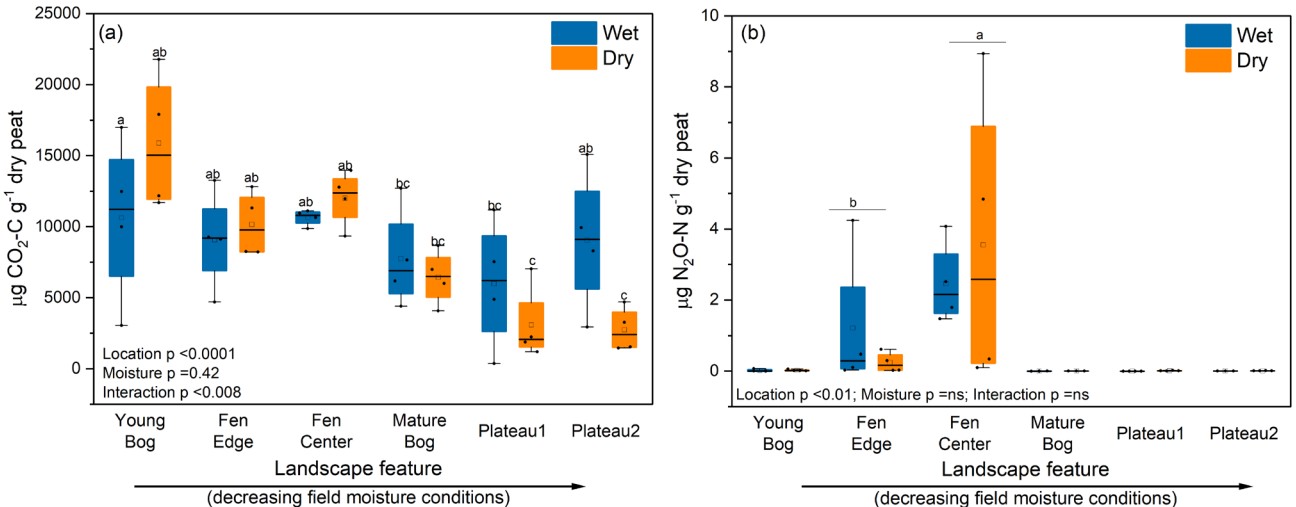



**Figure 3.** Cumulative $CO_2$ (a) and $N_2O$ (b) production by moisture treatment and landscape feature ordered from high to low *in situ* peat moisture. Wet treatments were incubated at field moisture conditions, and dry treatments were incubated under gradual drying. ANOVA p-values are shown for each factor and their interaction. Horizontal lines show the median (n=4) and boxes show the 25th and 75th percentiles. Means that do not share a same letter are significantly different.

Further supporting this site-dependent effect of drying on $CO_2$ production, a decrease in % peat moisture was strongly negatively correlated with increasing $CO_2$ and the strength of this relationship was strongest in the wettest sites ($r^2 > 0.8$), declining as *in situ* moisture also declined (Fig. 4). We also found a strong, positive correlation between field moisture and the difference in cumulative $CO_2$ emissions between treatments at the same site (Fig. 4). At initial peat moisture above 90%, gradual drying increased cumulative $CO_2$ emissions− conversely, below 90%, drying decreased cumulative $CO_2$. This suggests that a threshold exists for optimum peat moisture that affects the trajectory of microbial respiration responses to short-term gradual drying.

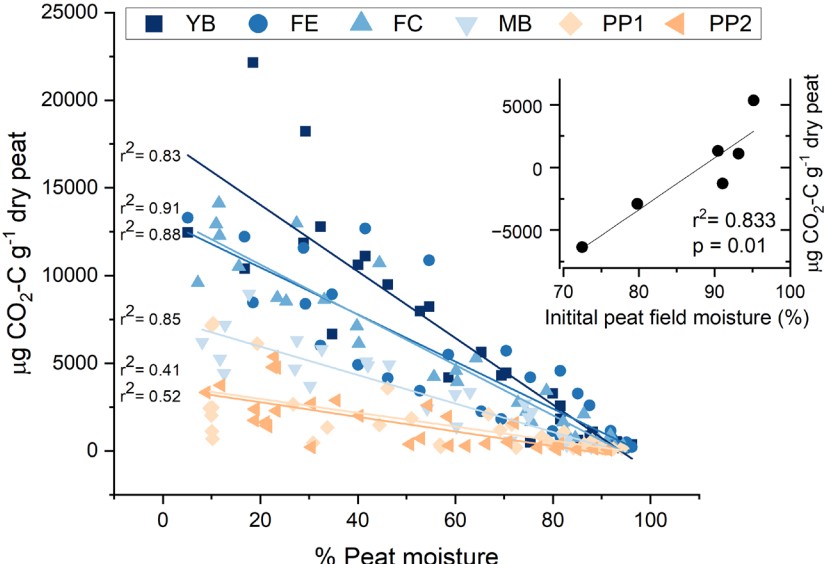

**Figure 4.** Linear regression of $CO_2$ production and peat moisture. Points are individual peat samples measured at 8 time periods while exposed to drying for 2 weeks. Landscape features are indicated by different colors and symbols in order of sites with the wettest to driest *in situ* field moisture conditions: young bog (YB), fen edge (FE), fen center (FC), mature bog (MB), peat plateau 1 (PP1), and peat plateau 2 (PP2). The inset is a regression of the mean initial peat field moisture and the absolute change in total $CO_2$-C after 2 weeks of gradual drying.

The site differences in total $CO_2$ production reflect what was observed in $CO_2$ production over time where, as the *in situ* peat moisture of the landscape feature became progressively drier, a negative impact of drying on $CO_2$ production emerged



(Fig. S3), occurring relatively early (within the first week) in the peat plateaus (p <0.005). Generally, $CO_2$ production increased linearly over time, regardless of the moisture treatment (Fig. S3). The PP2 (fen transect) was the exception, where after 1.5 weeks we no longer observed significant increases in net $CO_2$ production.

Our short-term experimental drying would allow for increased aerobic respiration in initially inundated or near-saturated environments, like the YB and FE. However, all our landscape features had similar moisture after drying. Thus, the wetter features (YB, FE) that responded positively to drying were likely influenced by other microbial metabolic drivers that coincide with moisture. For example, more microbial available C is more likely to accumulate in wetter sites than in the drier peat plateaus, where aerobic metabolism would be less constrained, reducing readily available C supply (Treat et al., 2014). Drying the peat plateaus may have thus suppressed $CO_2$ production because C became limited faster than at the wetter sites. Thus, it was not necessarily moisture conditions during the incubation driving our opposing responses, but rather the initial moisture conditions and covarying factors such as C availability and reactivity specific to each landscape feature.

Some studies suggest that drier, more oxic permafrost peatlands contribute more to the positive carbon-climate feedback effect than anoxic soils because of the higher rates of $CO_2$ production (Schädel et al., 2016; Schuur et al., 2015; but see Knoblauch et al., 2018). Our data suggest it is more nuanced than this where, all else being equal, lower moisture by itself may reduce $CO_2$ in less degraded peat plateaus. At the same time, due to enhanced evapotranspiration, the edges of bogs and fens which form with thaw and the collapse of peat plateaus, are expected to experience drying first, which our results imply may increase $CO_2$ emissions from those sites. While we cannot account for the influence of vegetation or anoxic conditions that could enhance methane production, the potential outflux of C associated with hydrological changes may thus initially only be amplified from surface peat within permafrost collapse features, and not from more stable permafrost plateaus.

### 3.3 Respired Peat $\delta^{13}C$-$CO_2$

We observed a consistent and significant drying-induced trend in $\delta^{13}C$-$CO_2$ (Fig. 5). Within each landscape feature, gradual drying produced cumulative $CO_2$ with relatively higher $\delta^{13}C$ than the field-moist peat, although differences in $\delta^{13}C$ for the peat plateaus were insignificant.

The substantially higher $\delta^{13}C$-$CO_2$ with drying implies that the C sources being respired change systematically in response to drying. One potential explanation is that, under drier conditions, decomposition shifts away from *sphagnum*-derived peat that can be relatively $\delta^{13}C$-depleted due to its symbiosis with methanotrophs (Kip et al., 2010). However, the mechanism leading to a possible shift away from sphagnum decomposition towards other more enriched-$^{13}C$ sources is unclear.

More generally, $\delta^{13}C$-$CO_2$ enrichment could signal a shift towards respiration of more decomposed peat under drying. Typically, more decomposed C is expected to be enriched in $^{13}C$ (Agren et al., 1996) due to selective microbial respiration of $^{12}C$. Under high moisture, especially near saturation, metabolism of highly decomposed peat may be energetically





unfavorable. Improved aeration of the dried peat would lead to more oxidizing conditions, allowing for the mineralization of more decomposed C with greater $\delta^{13}C$ enrichment (Alewell et al., 2011). This might explain why we see less isotopic
differentiation in the peat plateaus where initial moisture was less limiting to aerobic decomposition. Previous studies have also shown that older, more degraded peat experienced greater degradation under reduced peat moisture conditions (Hardie et al., 2011). Overall, the isotopic data imply that changes in $CO_2$ production with drying are likely associated with a shift from decomposition of fresh plant-derived substrates, including *sphagnum*, to more degraded peat. Future incubations involving $^{14}C$ measurements could indicate whether this also entails a shift to the decomposition of older peat C pools.


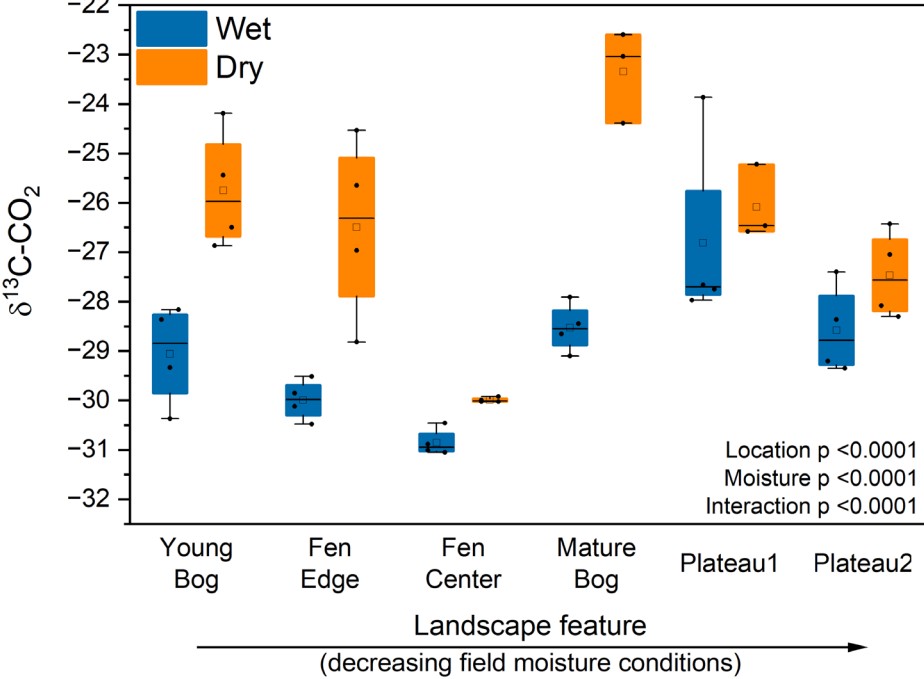

**Figure 5**. Cumulative $\delta^{13}C$-$CO_2$ by landscape feature and moisture treatment. Landscape features are ordered from high to low average *in situ* peat moisture. Wet moisture treatments were incubated at field moisture conditions and dry moisture treatments were incubated under gradual drying. Horizontal lines show the median (n=4) and boxes show the 25th and 75th
percentiles. Means that do not share a same letter are significantly different.

## 4 Conclusion

As permafrost thaws, changes to hydrologic conditions and atmospheric temperatures lead to moisture loss from peat in some circumstances, influencing microbial $CO_2$ and $N_2O$ production. We establish the importance of initial peat moisture on



the effect that shot-term gradual drying imparts on cumulative $CO_2$ emissions. Respiration is enhanced by drying in relatively wet sites and depressed by drying in relatively dry sites. Thus, as some permafrost systems undergo drying following thaw, the $CO_2$ response and subsequent loss of peat C may depend on antecedent moisture conditions, whereas $N_2O$ emissions may be largely controlled by total N availability. Furthermore, gradual drying increased the $^{13}C$ respiration, suggesting a shift to more decomposed, potentially older C mineralized with peat drying, implying the loss of previously

stable peat C reservoirs.

**Author Contribution**

AS helped conceive the experimental approach, was the primary person conducting the experiment, lab and data analyses, wrote the first draft of the manuscript, and contributed to subsequent drafts; CMK helped conceive the experimental

approach, and contributed to funding, data interpretations, and writing of all subsequent drafts; MSB helped design the experimental approach, conducted some of the experiment, and contributed to the writing; DO and CS collected the samples and data of the field sites, and contributed to the developed of the experiment and writing; RM contributed to the developed of the ideas for the experiment, laboratory analyses, and writing; PMJD contributed to funding, conceiving the experimental approach, data interpretations, and writing of all subsequent drafts.


**Data Availability**

All relevant data to this study can be accessed at from "Data collected from Lutose, Alberta surface peat permafrost samples", https://doi.org/10.5683/SP3/J6YIEJ, Borealis, V2.

**Supplemental Material**

The supplemental material related to this article is available online at: Xxxxxx

**Conflict of Interest**

The authors have no conflict of interest related to this research to declare.


**Acknowledgments**

We are grateful for the financial support for this research provided by the McGill Sustainability Systems Initiative Ideas Fund. We thank L. Galantini for help with gas analyses and Thi Hao Bui for help with stable isotope measurements.




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
