# Peer review of "Gradual drying of permafrost peat decreases carbon dioxide production in drier peat plateaus but not in wetter fens and bogs"

_EGUsphere, 2024_

## Author Response (AR1)

*We are grateful for the positive reviews and the opportunity to improve our manuscript. We have provided responses to the reviewers in red below and uploaded an updated manuscript file, along with a version with the tracked changes to easily see where revisions were made.*

**GENERAL COMMENTS**

This manuscript reports $CO_2$ and $N_2O$ production rates from a drying experiment with permafrost peatlands samples representing different thawing stages and field moisture contents. Permafrost thaw causes hydrological changes in both directions: it can cause either increased wetness or improved drainage. These changes in hydrology will impact greatly the soil GHG budget. The effect of increased wetness has been studied much more than the effect of drying, partly because the previous one is much easier to achieve. The drying approach chosen here is simple but effective, I really like it. Overall, this is a nice and compact, carefully planned and conducted study with clear results: $N_2O$ production from nutrient-rich sites with little moisture effect, and differential moisture effect depending on the initial moisture content and carbon quality. The experimental and statistical methods are suitable for the goals of the experiment, report is well written, all figures and tables are of a good quality and relevant, and the conclusions are well supported by the data. I have only minor suggestions, listed below.

*Thank you for your supportive review and helpful suggestions to improve our manuscript. The majority of the comments pointed to the need to clarify our wording and extrapolate on some results to give a bigger picture of the carbon and nitrogen sources and fluxes across our study site. We have made adjustments in response, as outlined below per comment.*

MINOR COMMENTS

line 60: With regards to methane, you should acknowledge that well-drained peatlands are known for their capacity to consume atmospheric $CH_4$ (Voigt et al. 2017, Voigt et al. 2024). I do not see it as a serious shortcoming that this flux was not measured here, but it would be good to mention this for a complete picture about peatland GHG budget.

Voigt, C. Marushchak, ME. Mastepanov, M. Lamprecht, RE. Christensen, TR. Dorodnikov, M. Jackowicz-Korczynski, M. Lindgren, A. Lohila, A. Nykänen, H. Oinonen, M. Oksanen, T. Palonen, V. Treat, CC. Martikainen, PJ. Biasi, C. (2019). Ecosystem carbon response of an Arctic peatland to simulated permafrost thaw. *Global change biology, 25* (5) , 1746-1764. 10.1111/gcb.14574.

Voigt, C., Virkkala, AM., Hould Gosselin, G. *et al.* Arctic soil methane sink increases with drier conditions and higher ecosystem respiration. *Nat. Clim. Chang.* **13**, 1095–1104 (2023). https://doi.org/10.1038/s41558-023-01785-3

*This is an interesting point! We agree that acknowledging the methane source and sink potential gives deeper insight into the complex and heterogenic nature of permafrost carbon fluxes. We have added a sentence noting this in the main text (L60: "Methane was not measured since experimental conditions were not anoxic, required for methanogenesis. However, we note that modeling and field experiments suggest that well-drained northern peatland soils uptake atmospheric methane, adding to the complexity*

*and heterogeneity of the C budget across a discontinuous permafrost peatland (Voigt et al., 2019; Voigt et al., 2023).".*

line 161 -> Since mineral N forms nitrate and ammonium are in a key role for N2O emissions, and you have actually measured these species, it would be good to discuss those results here a bit more. Was there any difference in mineral N content between landscape features? Did you observe increase or decrease in mineral N pools during the incubations? Also, you could comment the temporal pattern – was the N2O production rate stable throughout the experiment, or were there changes? On lines 164-166 you suggest that the N2O emissions in your experiment would origin from nitrification rather than denitrification. Is this in line with the lack of moisture effect, would not you then expect that the emissions would be lower in the dried peat?

*The reason we did not extrapolate on the results of our ammonium and nitrate was that there was no clear relationship between $N_2O$ production and inorganic N species across different sites or pre- vs. post-incubation. However, the %N in the solid material was significantly higher in the Fen Edge and Fen Center (1.8 and 2.44 respectively), where $N_2O$ was also detected, compared to the other sites. We understand that stating there was no impact of available nitrate on $N_2O$ production is important to give the full picture of the N dynamics in our system. Thus, we have included a brief reference to these results within this paragraph. Specifically, we added" The absence of $N_2O$ production from the drier peat plateau sites was expected as they also had the lowest nitrate and ammonium (Table S1) of all the sites. However, the bog sites had higher or similar inorganic N concentrations to the fens but also did not produce $N_2O$. Thus, it does not appear that there is a relationship between $N_2O$ production and inorganic N concentrations."; We further included the nitrate concentrations associated with the fen sites directly within the text (L169), in addition to the existing table. All ammonium and nitrate values can be found in our publicly available data: https://doi.org/10.5683/SP3/J6YIEJ .*

*We have also added a brief statement on the temporal trends (L164-165) (Fig. S2): Changes in cumulative $N_2O$ over time from the fen center peat generally increased linearly, whereas $N_2O$ production from the fen edge occurred exclusively during the last two sampling times from the wet control samples (Fig. S2)."*

*Regarding the discussion on the potential source of $N_2O$, we meant to suggest that from the previous Voigt 2020 review cited, inhibited nitrification in wetter conditions would reduce nitrate availability which might explain why they found lower $N_2O$ in wetlands. We did not mean to imply our $N_2O$ production was from nitrification. While possible, as you suggest, if true we would expect a drying effect to increase $N_2O$ production, which we did not observe. We have made some small changes to the wording to clarify this point.*

lines 171-174: While I do agree with this, it is important to acknowledge that by excluding the plant N uptake in the incubations, you enhance the N availability for microbes. Fen sites often have high productivity, when the plant cover is undisturbed, the plants will most likely take up most of the mineralized N. Please, acknowledge this in the discussion. However, your results are very relevant for the cold season and shoulder season when plant growth is low or absent.

*This is an excellent point! We have added your suggestion (L180-183): "However, we note that we cannot account for potential plant N uptake which could otherwise reduce available N for microbial $N_2O$ production, especially in highly productive fens. Nonetheless, during the cold or shoulder season or where plant growth is low or disturbed, these results support the emerging evidence that permafrost*

*ecosystems have N₂O hotspots that should be accounted for in estimating the global warming potential of GHG production."*

lines 185: To me this expression sounds a bit complicated, how about "% peat moisture was negatively correlated with CO2 production" or "a decrease in % peat moisture was associated with increasing CO2 production"

*In an effort to increase the clarity and concision of the writing, we have acted on the reviewer's suggestion and changed the wording of "a decrease in % peat moisture was strongly negatively correlated with increasing CO₂" to "a decrease in % peat moisture was associated with increasing CO₂ production."*

lines 187-191: Here, all the data series, independent on the landscape feature, seem to extend up to 100% peat moisture, although on row 142 above you say that the original field moisture content was varying between 73-95%. The makes me wonder if you always refer to %H2O from FW with "% Peat moisture" or do you sometimes mean the moisture content relative to the original field moisture content? Please clarify this throughout the MS, it seems very important for the interpretation of the results.

*The figure refers to peat moisture content during the incubation period, including values from within the first few days at the start of drying. Thus, some of the points still contain very wet samples and on the figure, these are close to 100% but not quite. However, this comment made us notice that some of the peat plateau moisture values were too high on the figure and we went back and checked the data that was plotted and found a small error in the plotted data. This was corrected with a revised figure but the revised figure and r-square values were only marginally impacted (see revised figure below). Finally, in all cases throughout the manuscript, we refer to %peat moisture as the absolute value measured from a given sample (so not relative to original moisture content). We have tried to clarify this throughout the text and in the figure captions.*

[Figure]

lines 201-202: Does this mean that the rate was stable = the CO2 concentration was increasing linearly? Please clarify.

*Yes, precisely. The cumulative $CO_2$ concentration increased linearly within the sealed incubation environment (for most samples). Thus, the $CO_2$ production rate was constant, or at least pretty consistent, over the two-week incubation period. We have reworded this section to clarify.*

line 222. I am curious if you observed any temporal trend in the wet treatment?

*We did not observe any temporal trend in the wet treatment. The average $\delta^{13}C$-$CO_2$ at the start and post-incubation at each site did not vary significantly (OB: p=0.7583; YB: p=0.6351; PP1: p=0.3245; FC: p=0.05727; FE: p=0.7524; PP2: p=0.1725). For the Fen Center, which has the lowest p-value for $\delta^{13}C$ values between the two time points, the average $\delta^{13}C$ difference in value was ~0.46, with a standard deviation of ~0.4. This is much lower than the deviations we see post-incubation between analogous dry and wet samples. We have added a sentence stating this result so as to assure readers that no temporal effect, independent of drying, impacted the $\delta^{13}C$-$CO_2$ composition (L238).*

line 204-> It is not completely clear which result you are explaining here. Do you mean the lower respiration rate observed in the dry landscape features at low moisture levels? I believe you are on the right track in that this is related to peat quality and nutrient status, which is in turn affecting the site productivity. So, the contrast between ombrotrophic bog and minerotrophic fen. Do you find any support from your results on peat chemistry?

*Yes, that is partially what we meant. However, more so between the wetter Fen and Bog sites and the drier peat plateau sites. Because there were no differences in moisture among the sites at the end of the drying experiment, the opposing response to drying between the wetter and drier (plateaus) landscape features is likely related to something besides a direct effect of lower moisture during the incubation. We propose it might be lower concentrations of available C in the peat plateau sites due to more efficient/higher metabolism under their in-situ non-saturated conditions. We have more explicitly added which results this discussion refers to and hopefully clarified these ideas. While we do not have data on peat available C, the peat plateau inorganic N is lower compared to the other sites at the start of the incubation and we now note within the text that the more rapid decline in overall $CO_2$ over time for the peat plateaus further suggests a C (or maybe nutrient) limitation occurring sooner compared to the wetter sites.*

line 217-220: This sentence is not so easy to understand, please check if you could rephrase/split into two sentences it to make it clearer?

*We agree the second clause of the sentence made this hard to read and have reworded the sentence as: "While we cannot account for the influence of vegetation or anoxic conditions that could promote methane production, our results suggest that hydrological changes may initially amplify surface peat carbon emissions primarily from permafrost collapse features, rather than from more stable permafrost plateaus."*

RC2:

General comments:

This article focusses on carbon dioxide and nitrous oxide production in permafrost affected peatlands, in particular on the way respiration will be affected by drying/wetting of the peat following permafrost thaw. The evidence is drawn from an elegant incubation experiment that

dries the peat samples without interfering with the incubation. The manuscript is very well written and the results are clearly presented. These types of incubation studies are key to increase our understanding of the responses of peatlands to changing environmental conditions. This study also measures $N_2O$ production, which adds another element of novelty. Additionally, this particular study focussing on changing moisture conditions is extremely useful for developing peatland models – as the relationship of decomposition with moisture is less well established in these models but it is an essential part if we want to capture responses of peatlands to future climate change.

Small comments:

- 1 Gradual drying of permafrost peat decreases carbon dioxide **production** in drier peat plateaus but not in wetter fens and bogs

- 19 Only the fen sites, which had high moisture and % total N, produced $N_2O$ (0.06−6.7 µg $N_2O$-N $g^{-1}$ dry peat) but t**he production of $N_2O$ was** unaffected by the drying treatments.

- 56 We conducted an incubation to determine the relationship between experimental short-term gradual drying and the magnitude of $CO_2$ and $N_2O$ gas production across a northern peatland thaw gradient and how drying affects the **respired** $CO_2$ isotopic **composition**.

- 59 We interpreted this isotopic **composition** as a signal of the C source being respired under drier conditions.

- 61 2) a decrease in moisture conditions w**ill lead to the production of more $^{13}$C-enriched $CO_2$**, possibly reflecting the metabolism of **more** decomposed peat. (I think this is a typo – you mean more decomposed peat?)

- 232 Under high moisture, especially near saturation, metabolism of highly decomposed peat may be energetically unfavorable. Improved aeration of the dried peat would lead to more oxidizing conditions, allowing for the mineralization of more decomposed C with greater δ13C **values** (Alewell et al., 2011).

*All of the reviewer's comments and suggestions have been incorporated into the text. We agree that the changes in wording clarify the meanings of the relevant sentences and better represent the study's methodology.*